# Dynamic Navigation System vs. Free-Hand Approach in Microsurgical and Non-Surgical Endodontics: A Systematic Review and Meta-Analysis of Experimental Studies

**DOI:** 10.3390/jcm12185845

**Published:** 2023-09-08

**Authors:** Elina Mekhdieva, Massimo Del Fabbro, Mario Alovisi, Nicola Scotti, Allegra Comba, Elio Berutti, Damiano Pasqualini

**Affiliations:** 1Department of Surgical Sciences, Dental School, Endodontics and Operative Dentistry, University of Turin, 10124 Torino, Italy; mario.alovisi@unito.it (M.A.); nicola.scotti@unito.it (N.S.); allegra.comba@unito.it (A.C.); elio.berutti@unito.it (E.B.); damiano.pasqualini@unito.it (D.P.); 2Department of Biomedical, Surgical and Dental Sciences, University of Milan, 20133 Milan, Italy; massimo.delfabbro@unimi.it; 3Fondazione IRCCS Ca’ Granda Ospedale Maggiore Policlinico, 20122 Milan, Italy

**Keywords:** dynamic navigation system, microsurgery, endodontics, systematic review, meta-analysis

## Abstract

(1) Background: A Dynamic Navigation System (DNS) is an innovative tool that facilitates the management of complex endodontic cases. Despite the number of advantages and limitations of this approach, there is no evidence-based information about its efficiency in comparison with that of the traditional method in endodontics. (2) Objectives: We aimed to explore any beneficial effects of the DNS and compare the outcomes of DNS vs. free-hand (FH) approaches for non-surgical and microsurgical endodontics. (3) Methods: A literature search was conducted in August 2023 to identify randomized, experimental, non-surgical, and microsurgical endodontic studies that compared the DNS with FH approaches. The procedural time (ΔT, s), substance loss (ΔV, mm^3^), angular deviation (ΔAD, °), coronal/platform linear deviation (ΔLD_C, mm), and apical linear deviation (ΔLD_A, mm) were recorded and analyzed. Quality and risk of bias assessments were conducted according to the Quality Assessment Tool For In Vitro Studies. A meta-analysis was performed using mean difference and standard deviation for each outcome, and heterogeneity (*I*^2^) was estimated. *p* < 0.05 was considered significant. (4) Results: One-hundred and forty-six studies were identified following duplicate removal, and nine were included in the systematic review and meta-analysis. The overall risk of bias was classified as low. The DNS was found to be more accurate and efficient than the FH approach was, resulting in a significantly shorter operation time (*p* < 0.00001) and less angular (*p* ≤ 0.0001) and linear deviation (*p* ≤ 0.01). For substance loss, the advantage of the DNS was significant only for microsurgery (*p* = 0.65, and *p* < 0.005, for non-surgical and microsurgical procedures, respectively). A reduced risk of iatrogenic failure using the DNS was observed for both expert and novice operators. (5) Conclusions: The DNS appears beneficial for non-surgical and microsurgical endodontics, regardless of the operator’s experience. However, appropriate training and experience are necessary to access the full advantages offered by the DNS.

## 1. Introduction

The main methods of pulp disease therapy are aimed at preventing periodontal complications. Non-surgical types of treatment are performed through gaining coronal access into the system of root canals. In the few clinical cases when direct coronal access is not performable due to risk of tissue loss, coronal blocks, or other indications, a microsurgical type of endodontic treatment could be the chosen treatment. In this case, access to root apexes is gained retrograde through the jaw bone structure. In both positions, tissue preservation (dentine and bone) is critical for a better prognosis and long-term outcome.

A Dynamic Navigation System (DNS) is an innovative tool that facilitates the management of complex endodontic cases (for example, accessing calcified canals, managing complicated anatomy, endodontic retreatment, and microsurgery) [1,2,3,4,5,6]. The first study to assess the use of the DNS was initially dedicated to accurate implant placement [7] after improved guided navigation during endodontic [8] and microsurgical procedures [9]. The DNS integrates spatial positioning technology and preoperative cone-beam computed tomographic (CBCT) imaging using cameras as an optical triangulation tracking system controlled with computer software [10,11,12]). The cameras and motion-tracking devices are attached to the dental handpiece and patient, respectively. The DNS device guides drilling at the target position according to a preoperatively planned angle, pathway and depth of the endodontic access cavities, and it allows real-time monitoring [13,14,15,16]. As such, the DNS enables more accurate and safer endodontic cavity access than conventional free-hand (FH) techniques do [17]. Furthermore, the DNS demonstrates improved usability in posterior regions compared with that of static guides [18,19], and real-time tracking enables immediate adjustment of the drilling path [20,21] These features may help prevent intraoperative complications (such as overextended access cavities, dentin loss, crown and root perforations, missed root canals, the fracture of root canal instruments during canal preparation, or the weakening of the coronal structure), increase the efficacy, and reduce the procedural time [17,22,23,24,25]

Despite the undeniable practical advantages of the DNS, there are associated limitations [26], and over-dependence on technology is a cause for concern. Most notably, the additional training time required for the DNS, which involves complicated instructions, the need for a learning curve, and additional preoperative time needed for software setting; scanning; and virtual designing and tracing, in addition to intraoperative calibration and checking as well as the difficulty in maintaining the visibility of the system display during clinical procedures often leads to the adoption of an FH approach instead [27].

Previous reviews examining the DNS vs. FH approaches have reported high data heterogeneity and low-level evidence from in vitro studies, thereby confounding any subsequent meta-analysis [28,29]. However, a qualitative analysis has reported the benefits of the DNS compared with those of FH approaches.

This systematic review and meta-analysis aimed to compare the accuracy and efficacy of DNS with FH approaches for non-surgical and microsurgical endodontic procedures, testing the null hypothesis (H_0_) that there would be no differences between the DNS and FH approaches.

## 2. Methods

This study was written in accordance with Preferred Reporting Items for Systematic Reviews and Meta-Analyses (PRISMA) guidelines (available at Appendix A) [30,31,32,33,34] and registered with the International Prospective Register of Systematic Reviews (PROSPERO) a priori; ID: CRD42022348725, https://www.crd.york.ac.uk/prospero/display_record.php?RecordID=348725 (accessed on 4 August 2022).

### 2.1. Eligibility Criteria

This analysis considered all experimental studies that evaluated endodontic procedures (both non-surgical and microsurgical) on human teeth (naturally extracted, cadaver, artificially 3D-printed ones) using the dynamic navigation system (DNS) in comparison with a free-hand (FH) approach. Review question: Does the DNS increase the accuracy and efficacy of performing endodontic and microsurgical procedures vs. an FH approach?

The methodology was carried out according to the Cochrane PICOS formula [35] This is defined as follows: Population: endodontically and microsurgically treated teeth with natural and artificial calcifications (human, extracted, 3D-printed teeth, and cadaver ones); Intervention: endodontic and microsurgical procedures performed using the DNS; Comparison: endodontic and microsurgical procedures performed using an FH approach; Outcomes: procedural time (s), substance volume loss (mm^3^), angular deviation (degrees), coronal (platform) linear deviation (mm), and apical linear deviation (mm); Study design: in vitro randomized experimental studies.

### 2.2. Information Sources and Search Strategy

A systematic literature search of PubMed, Cochrane Library, Wiley Online Library, and Scopus electronic databases was conducted in August 2023 to include all randomized experimental (in vitro) studies that evaluated endodontic procedures (both non-surgical and microsurgical) utilizing the DNS (test) in comparison with FH (control) approaches on human teeth. Studies were excluded if they did not include an FH control group. The search was limited to articles published in journals over the last five years, without any other restriction. The search terms used were the following: “Dynamic navigation” and “Endodontics”, which are also MeSH terms, and they were combined using the Boolean operator AND. They were entered in the fields for title, abstract, and key words. The search query was adapted to each database.

In addition, the references cited in each selected study were screened, and any that met the inclusion criteria were added to the list.

### 2.3. Selection and Data Collection Process

The literature search, study screening, and data extraction were independently performed by two reviewers (EM and MDF). Any disagreements were resolved through a discussion. More specifically, the studies initially retrieved through electronic plus manual searches were added to a reference manager software program (Zotero 6.0.26), which was also used to exclude duplicate articles. The titles and abstracts of the remaining studies were then screened to form a list of eligible studies. The full text of all eligible studies was obtained and assessed to make sure that the studies met the inclusion criteria. All studies excluded at this stage were listed in a separate table, indicating the reason for exclusion. The included studies underwent data extraction for the subsequent steps of the review (qualitative and quantitative synthesis).

The following outcomes were extracted from the included studies and expressed as the difference (Δ) between the DNS and FH groups: procedural time (ΔT, s), substance volume loss (ΔV, mm^3^), angular deviation (ΔAD, °), coronal (platform) linear deviation (ΔLD_C, mm), and apical linear deviation (ΔLD_A, mm). The following information was collected in addition to the outcome measures listed above: author names, study title, year of publication, country, study design, quality and quantity of prepared teeth, techniques (DNS and FH), and the number and experience of operators performing the study. In the event of missing data, the study’s authors were contacted via email. The respective study was excluded if no response was received after three attempts within three weeks.

### 2.4. Data Items and Statistical Analysis

Separate meta-analyses were performed for each outcome measure (procedural time, substance loss, angular deviation, coronal (platform) linear deviation, and apical linear deviation) using the software Review Manager 5.4 (Cochrane Collaboration, 2020). Data heterogeneity was evaluated using funnel plots. In the instance of there being no significant heterogeneity (defined as *I*^2^ < 60%, *p* > 0.05), mean differences for continuous data were combined using fixed-effects models. A random-effect model was adopted when *I*^2^ > 60%, and *p* < 0.05.

### 2.5. Risk of Bias Assessment

The Quality Assessment Tool For In Vitro Studies (QUIN Tool) was used to assess the risk of bias of the included studies [36]). Two reviewers scored each item from each study as adequately specified (score = 2), inadequately specified (score = 1), not specified (score = 0), or not applicable (exclude criteria from the calculation). The risk of bias was then assessed using twelve criteria and this formula: Final Score = (Total score × 100)/(2 × number of criteria applicable). The final scores were used to grade the in vitro studies as having a low (>70%), medium (50–70%), or high (<50%) risk of bias. Any disagreements were resolved through discussion and consultation with a third reviewer.

## 3. Results

### 3.1. Study Selection

The literature search yielded 146 articles following the removal of duplicates. The subsequent screening of the titles and abstracts led to the exclusion of 124 publications which were not relevant to this analysis. The remaining 22 studies underwent full-text evaluation, which led to the exclusion of a further 13 articles that did not meet the inclusion criteria. A flowchart of the selection process is shown in Figure 1. Table 1 lists the 13 studies [1,3,12,14,15,20,22,25,37,38,39,40] excluded following full-text evaluation, with the reasons for exclusion. No additional articles were added from the reference lists of the selected studies, leading to a total selection of nine articles for this systematic review [11,13,27,41,42,43,44,45,46].

### 3.2. Study Characteristics

A summary of the included studies is presented in Table 2. Two of the studies reported all five outcome measures evaluated in this review [43,45], three studies reported four of the five parameters [13,41,44], two studies reported three parameters [42,46]), and the remaining two studies reported two parameters [11,27].

Five of the nine studies used natural, extracted human teeth (with calcified canals, or from cadavers) [13,41,43,44,46], and four used artificial teeth made via 3D printing or made from resin (with the artificial simulation of canal obliteration) [11,27,45]. Six studies were dedicated to non-surgical treatment [11,13,27,42,43,46], and three carried out microsurgical procedures [41,44,45]. One study [42] (only included molar teeth, while the rest examined mainly anterior, single-rooted teeth.

The subgroup sizes ranged from 10 [42] to 36 [27] teeth. The procedures in three of the studies were performed by a single experienced endodontist [41,43,46] while one study was conducted by two experienced operators [42]). Four studies were each performed by a pair of doctors with different skill levels (one expert and one novice) [13,27,44,45], and the final study was conducted by a novice operator (second-year resident) [11].

Generally, nine selected studies reported that the DNS was more accurate and more efficient than an FH approach for access preparation, canal location, and fiber post removal. Compared with an FH approach, the DNS resulted in significantly reduced substance loss and a shorter operation time. The DNS was also considered to minimize the potential risk of the iatrogenic weakening of critical portions of the crown and reduce negative influences on shaping procedures [11]). According to one study that evaluated the impact of skill level, the results obtained using the DNS were independent of the operator’s experience [27]. In that study, less experienced operators accomplished more minimally invasive access cavity preparations than those prepared by more experienced operators did [27]. Furthermore, several studies identified accuracy and efficiency as advantages of the DNS for conducting minimally invasive osteotomy and root-end resection for microsurgical procedures [8,45,46].

### 3.3. Risk of Bias in Studies

The criteria of the Quality Assessment Tool For In Vitro Studies (QUIN Tool) are reported in Table 3 [9]. All nine included studies were classified as having a low risk of bias (Table 4).

### 3.4. Results of Individual Studies and Syntheses: Meta-Analysis

All nine studies were included in a random-effect meta-analysis on procedural time [11,13,22,41,42,43,44,45,46]. The duration of the endodontic treatment when using the DNS ranged from 11.5 to 241.8 s for access preparation and from 257.0 to 550.0 s for endodontic microsurgery. According to Figure 2, which shows a Forest plot of mean difference in procedural time and substance loss, the DNS significantly minimized operative procedural time in comparison with that of the FH approach overall (*p* < 0.00001) and in the non-surgical and microsurgical subgroups (Figure 2A). A high heterogeneity level was identified in both subgroups (*I*^2^ = 95% and 96%, respectively). The effect was comparable across the two subgroups and appeared independent of operator’s experience in the two microsurgical studies that were stratified by experience. It is noteworthy that almost all the included studies defined the procedural time as the in-chair time only, excluding any preoperative time required for CBCT scanning, digital planning, sensor calibration, and other procedures not required for an FH approach. Due to the nature of the studies that included expert and novice operator subgroups, it was not possible to compare the procedural time spent by the expert and novice operators [11,44,45]. In addition, the study of Connert et al. (2021) [27] did not report data that were stratified by experience level.

Three non-surgical studies and two subgroups of one microsurgical study were included in an analysis of substance loss (Figure 2B, Connert et al., 2021 [27], Jain et al., 2020 [11]; Janabi et al., 2021 [43]; Tang et al., 2022 [45]). Two studies in the non-surgical group identified an advantage of using the DNS (Connert et al., 2021 [27], Jain et al., 2020 [11]), while one (Janabi et al., 2021 [43]) reported reduced substance loss in the FH group. All three studies had an equal weight close to 20%. Overall, the effect was not statistically significant (*p* = 0.65, *I*^2^ = 99%). In the microsurgical group, separate novice and expert data both favored the DNS, with significant difference between the subgroups (*p* = 0.005, *I*^2^ = 66%).

Angular deviation was reported in six studies. According to Figure 3, which shows a Forest plot of mean difference in angular deviation, the DNS demonstrated significantly less angular deviation compared with an FH approach to non-surgical (*p* = 0.0001, Figure 3A) and microsurgical procedures (*p* < 0.00001, Figure 3B). High heterogeneity was detected in both groups (*I*^2^ = 86% and 82%, respectively).

Following Figure 4, which shows a Forest plot of mean difference in coronal/platform linear deviation, the use of the DNS also resulted in significantly less coronal/platform linear deviation than the FH approach in the non-surgical (*p* < 0.00001, Figure 4A) and microsurgical studies did (*p* = 0.01, Figure 4B). Although no heterogeneity was detected among the four non-surgical studies (*I*^2^ = 0%), high heterogeneity was identified in the surgical studies (*I*^2^ = 99%), primarily due to one study which reported a significantly greater benefit of the DNS when used by novice operators compared with experts (41).

Finally, according to Figure 5, which shows a Forest plot of mean difference in apical linear deviation, our analysis also identified a significant advantage of the DNS compared with the FH approach in apical linear deviation for both non-surgical (*p* = 0.0008; *I*^2^ = 0%) and microsurgical (*p* < 0.00001; *I*^2^ = 69%) procedures (Figure 5A, Figure 5B).

## 4. Discussion

This systematic review sought to investigate and compare the available evidence regarding the efficacy of the DNS and FH approaches for non-surgical and microsurgical endodontics. Although several DNS studies were identified, our study design and stringent inclusion criteria meant this review and meta-analysis was limited to nine in vitro studies. It is also notable that we were able to perform a meta-analysis precluded from previous systematic reviews. No previous systematic review [28,29,46] used the same eligibility criteria as those of this study. Zubizaretta et al. (2021) [46] performed a systematic review comparing the DNS vs. a static approach to guided endodontics. Jonaityte et al. (2022) [28] and Vasudevan et al. (2022) [29] did not perform meta-analyses on any outcome measures and did not include studies on endodontic microsurgery.

Time is an important consideration when planning a treatment and also to ensure a comfort relationship between the patient and doctor. This meta-analysis found that the DNS was associated with a significantly reduced procedural time (i.e., duration of the procedure) compared with that of the FH approach. It is noteworthy that almost all the included studies defined procedural time as the in-chair time only, excluding any preoperative time required for CBCT scanning, digital planning, sensor calibration, and other procedures not required for an FH approach. Due to the nature of the studies that included expert and novice operator subgroups, it was not possible to compare the procedural times spent by expert and novice operators [11,44,45]. In addition, the study of Connert et al. (2021) [27] did not report data stratified by experience level. Dianat et al. (2020) [13] underlined that the mean time required for locating canals was significantly reduced with the DNS vs. an FH approach (*p* < 0.05), and that the time required to prepare access cavities was significantly reduced for expert vs. novice operators (*p* < 0.05). The DNS was also associated with a decreased access preparation time (average of 4 min; maximum of 7 min) vs. an FH approach (average 7 min; maximum of 19 min). Jain et al. (2020) reported significantly faster access preparation with the DNS compared with that of the FH approach (2.2 vs. 7.06 min, *p* < 0.05), while Connert et al. (2021) [27] reported no significant difference in the mean procedural time between the DNS and FH approaches (193 vs. 195 s, *p* > 0.05). According to Janabi et al. (2021) [43], the average time taken for post removal with the FH approach was double that of the DNS (8.30 vs. 4.03 min, *p* < 0.05). This is supported by Martinho et al. (2022) [44], who reported that the time required for osteotomy and root-end resection using the FH approach was double that of the DNS approach, regardless of the operator’s experience. Finally, Adalmash et al. (2022) [41] also found that the DNS was associated with significantly less time vs. the FH approach for microsurgery procedures (*p* < 0.05). Only Tang et al. (2022) [45] reported the time spent on clinical assessment and surgical time separately, but did not detail how the beginning and end steps of each procedure corresponded to the recorded times. The final time point in the DNS approach should likely be represented by reaching the planned mark on a digital scan, while the corresponding time point for the FH approach should be when the canal orifice is free of calcification, or other steps, depending on the type of procedure. Thus, differences in the procedural time may be influenced by inter-study variation in the measurement of time parameters.

Jain et al. (2020) [11] reported significantly reduced mean substance loss with the DNS compared with that of the FH approach (27.2 vs. 40.7 mm^3^, *p* < 0.05). Data reported by Connert et al. (2021) [27] also favored the DNS vs. the FH approach in this respect (10.5 vs. 29.7 mm^3^, *p* < 0.001). These results were further supported by Janabi et al. (2021) [43], who demonstrated the significantly reduced volumetric loss of the tooth structure with the DNS vs. the FH approach, and Tang et al. (2022) [45], who compared the DNS vs. FH approaches for endodontic microsurgery by expert and novice operators.

Linear and apical deviations can be used to determine the precision and accuracy of a navigation approach. In this regard, Gambarini et al. (2020) [42] determined that the DNS approach was significantly more precise than the FH approach, with less angular and linear deviation (4.8° and 0.34 mm vs. 19.2° and 0.88 mm for DNS vs. FH, respectively). Data from Janabi et al. (2021) [43] concurred with this observation, reporting significantly less linear and apical deviation with the DNS vs. FH approaches (*p* < 0.05). This was further confirmed by Martinho et al. (2022) [44] who reported significantly higher accuracy using the DNS vs. FH approaches (*p* < 0.005) and verified that expert operators achieved a higher accuracy with the DNS than novice operators did. Finally, Dianat et al. (2020) [37] also observed significantly less deviation (*p* < 0.001) and angular deflection (*p* < 0.0001) with the DNS vs. FH approaches. According to Tang et al. (2022) [45], the DNS significantly reduced the gap in these indications, as compared with that of the FH group, regardless of the operator’s experience.

In summary, this systematic literature search and meta-analysis suggests that the DNS is associated with significantly reduced substance loss (in microsurgical procedures) and shorter operation times than those of an FH approach, while also minimizing the potential risk of the iatrogenic weakening of critical portions of the crown and reducing negative influences on shaping procedures. Overall, the data indicate that the DNS is a highly accurate and efficient tool for difficult clinical cases where extreme precision is required [11,13,28,37,41,42,43,45,46]. The DNS has a broad range of applications, including microsurgery, retreatment, primary treatment, the opening of the obliterated pulp chamber and canals, and the removal of fiber posts. Two studies that evaluated the impact of operators’ experience on the DNS-associated outcomes reported that novice operators accomplished more minimally invasive access cavity preparations comparable to those of experienced operators [42,45]. Although this suggests that the DNS outcomes may be independent of the operators’ experience, we would urge caution in this regard and suggest that further specifically designed studies are needed to precisely determine the skill and experience required to achieve optimal results with the DNS.

The key strength of this systematic review and meta-analysis is the strict eligibility criteria that were used to identify the comparative studies of the DNS and FH approaches, which minimized the risk of selection bias. The low overall risk of bias of the included studies further strengthens these analyses.

Given the relative limitations of this systematic review and meta-analysis, more studies are needed to fully determine the appropriate use of the DNS vs. FH approaches. Future studies should consider using standardized procedures for separated teeth groups and well-defined time measurements, while taking into account individual clinical case characteristics and operators’ training and expertise.

## 5. Conclusions

According to the results of this systematic review and meta-analysis, the DNS approach for non-surgical and microsurgical endodontic procedures appears to yield better results when compared with those of an FH approach in almost every subgroup examined. The benefits of the DNS range from a reduced operating time to a reduction of substance loss and a lessened probability of iatrogenic complications, and it also improves accuracy of the treatment. However, until recently, the use of the DNS was optional and often dependent on personal experience, training, and manual skills to derive the full advantages of the technique. With limited available evidence, there is a need for high-quality trials to fully determine the benefits offered by the DNS approach for non-surgical endodontics and microsurgery.

## Figures and Tables

**Figure 1 jcm-12-05845-f001:**
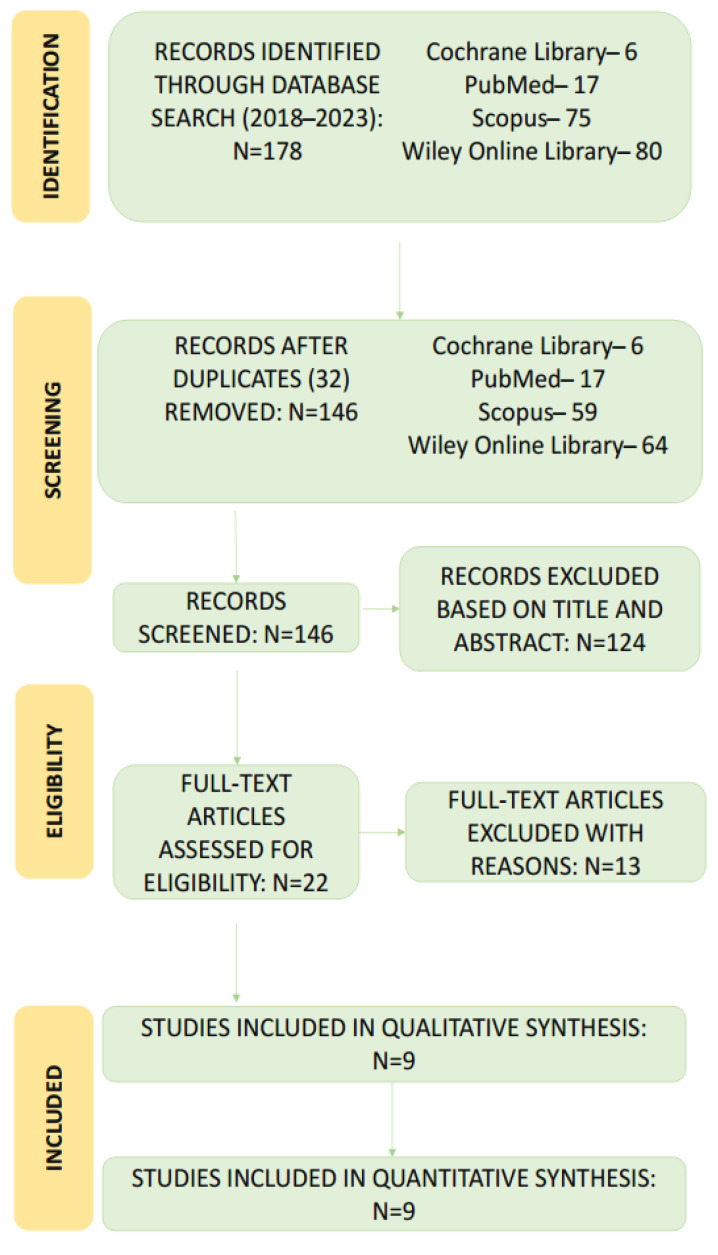
A Preferred Reporting Items for Systematic Reviews and Meta-analyses flowchart of search results and screening process.

**Figure 2 jcm-12-05845-f002:**
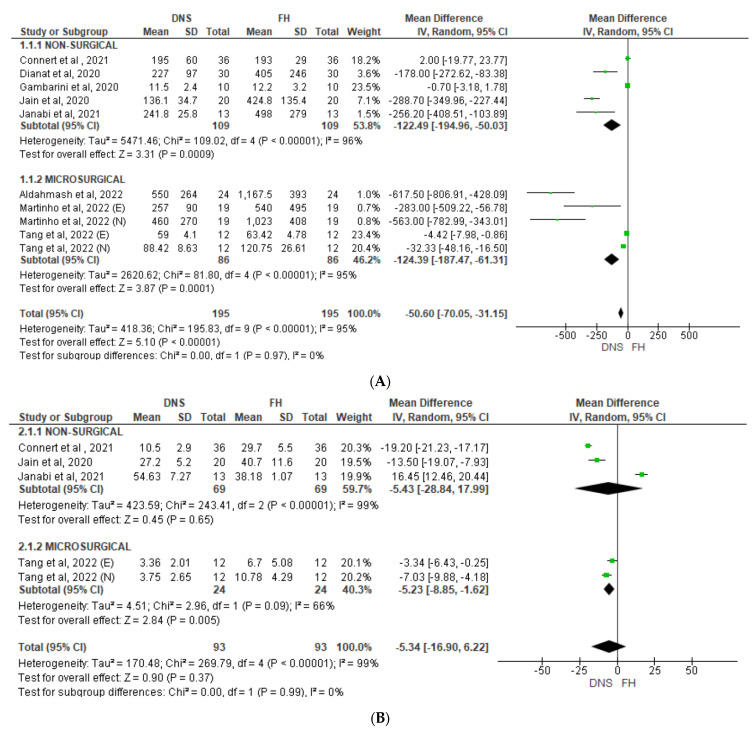
Forest plot of mean difference in (**A**) procedural time and (**B**) substance loss between DNS and FH approaches for non-surgical and microsurgical procedures. DNS, dynamic navigation system; FH, free-hand; E, experienced; N, novice [11,27,41,43,44,45].

**Figure 3 jcm-12-05845-f003:**
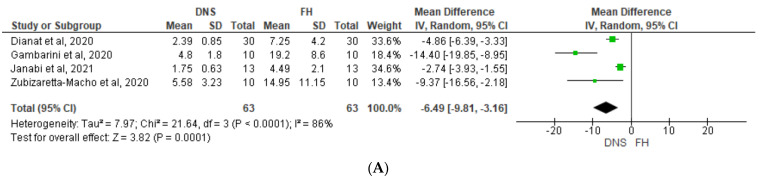
Forest plot of mean difference in angular deviation between DNS and FH approaches for (**A**) non-surgical and (**B**) microsurgical procedures. DNS, dynamic navigation system; FH, free-hand; E, experienced; N, novice [13,41,42,43,44,45,46].

**Figure 4 jcm-12-05845-f004:**
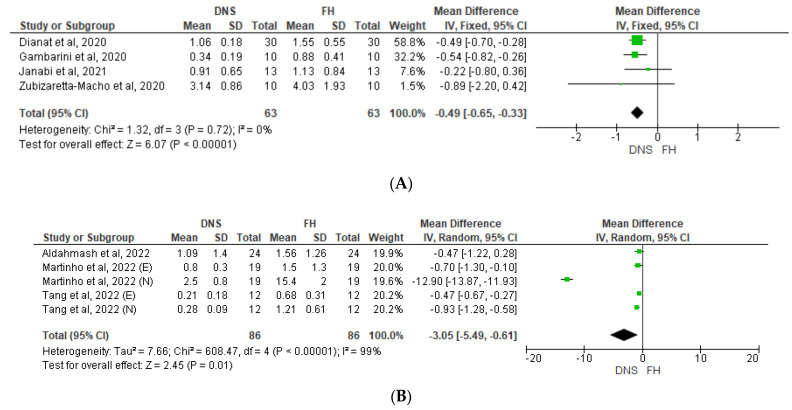
Forest plot of mean difference in coronal/platform linear deviation between DNS and FH approaches for (**A**) non-surgical and (**B**) microsurgical procedures. DNS, dynamic navigation system; FH, free-hand; E, experienced; N, novice [13,41,42,43,44,45,46].

**Figure 5 jcm-12-05845-f005:**
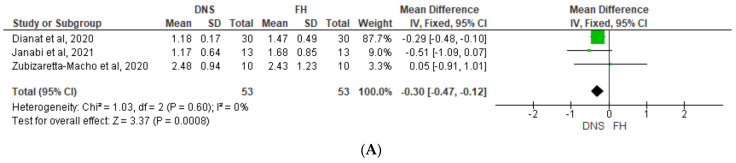
Forest plot of mean difference in apical linear deviation between DNS and FH approaches for (**A**) non-surgical and (**B**) microsurgical procedures. DNS, dynamic navigation system; FH, free-hand; E, experienced; N, novice [13,41,43,44,45,46].

**Table 1 jcm-12-05845-t001:** List of excluded studies (with reasons).

#	Authors	Study	Reason of Exclusion
1	Chong et al., 2019 [20]	Computer-aided dynamic navigation: a novel method for guided endodontics http://doi.org/10.3290/j.qi.a41921	No control (freehand) group
2	Torres et al., 2021 [25]	Dynamic navigation: a laboratory study on the accuracy and potential use of guided root canal treatment http://doi.org/10.1111/iej.13563	No control (freehand) group
3	Bardales-Alcocer et al., 2021 [1]	Endodontic retreatment using dynamic navigation http://doi.org/10.1016/j.joen.2021.03.0005	A case report
4	Leontiev et al., 2022 [14]	Dynamic navigation in endodontics: guided access cavity preparation by means of a miniaturized navigation system http://doi.org/10.3791/63687	No control (freehand) group
6	Dianat et al., 2021 [37]	Guided endodontic access in a maxillary molar using a dynamic navigation system http://doi.org/10.1016/j.joen.2020.09.019	A case report
7	Jain et al., 2020 [3]	3-Dimensional accuracy of dynamic navigation technology in locating calcified canals http://doi.org/10.1016/j.joen.2020.03.014	No control (freehand) group
8	Simon et al., 2019 [12]	Computer-controlled CO₂ laser ablation system for cone-beam computed tomography and digital image guided endodontic access: a pilot study http://doi.org/10.1016/j.joen.2021.06.004	No control (freehand) group
9	Liu et al., 2022 [15]	In vitro evaluation of positioning accuracy of trephine bur at different depths by dynamic navigation http://doi.org/10.19723/j.issn.1671-167X.2022.01.023	No control (freehand) group
10	Gambarini et al., 2019 [22]	Endodontic microsurgery using dynamic navigation system http://doi.org/10.1016/j.joen.2019.07.010	A case report
11	Chen et al., 2023 [38]	Clinical and radiological outcomes of dynamic navigation in endodontic microsurgery: a prospective study https://doi.org/10.1007/s00784-023-05152-6	A clinical study
12	Karim et al., 2023 [39]	Comparative Evaluation of a Dynamic Navigation System versus a Three-dimensional Microscope in Retrieving Separated Endodontic Files: An In Vitro Study https://doi.org/10.1016/j.joen.2023.06.014	The study is focused on retrieving broken rotary files
13	Martinho et al., 2023 [40]	Augmented Reality and 3-Dimensional Dynamic Navigation System Integration for Osteotomy and Root-end Resection https://doi.org/10.1016/j.joen.2023.07.007	No control (freehand) group

**Table 2 jcm-12-05845-t002:** Summary of characteristics of the included studies.

Study number	**1**	**2**	**3**	**4**	**5**	**6**	**7**	**8**	**9**
First author	Dianat et al. [13]	Jain et al. [11]	Connert et al. [27]	Gambarini et al. [42]	Janabi et al. [43]	Zubizarreta-Macho et al. [46]	Martinho et al. [44]	Aldahmash et al. [41]	Tang et al. [45]
Year	2020	2020	2021	2020	2021	2020	2022	2022	2022
Country	USA, Saudi Arabia	USA	Switzerland, Germany	Italy, Saudi Arabia, Russian Federation	USA	Spain	USA, Saudi Arabia	Saudi Arabia, USA	China
Study design	IN VITRO	IN VITRO	IN VITRO	IN VITRO	IN VITRO	IN VITRO	IN VITRO	IN VITRO	IN VITRO
Teeth	Extracted	3D printed	3D printed	Artificial (resin)	Extracted	Extracted	Cadavers’	Cadavers’	3D printed
Group	Anterior, premolars	Incisors	Anterior	Upper right first molars	Incisors, canines	Anterior	Anterior, posterior	Anterior, posterior	Anterior, posterior
DNS sample size	30	20	36	10	13	10	19	24	12
Freehand sample size	30	20	36	10	13	10	19	24	12
Operator skill level	E, N	N	E, N	E	E	E	E, N	E	E, N
Reported outcomes									
**Substance loss**	-	+	+	-	+	-	-	-	+
**Procedural time**	+	+	+	+	+	-	+	+	+
**Angular deviation**	+	-	-	+	+	+	+	+	+
**Linear deviation coronal**	+	-	-	+	+	+	+	+	+
**Linear deviation apical**	+	-	-	-	+	+	+	+	+

3D, three-dimensional; DNS, dynamic navigation system; E, experienced. N, novice.

**Table 3 jcm-12-05845-t003:** QUIN Tool criteria.

#	Criteria
1	Clearly stated aims/objectives
2	Detailed explanation of sample size calculation
3	Detailed explanation of sampling technique
4	Details of comparison group
5	Detailed explanation of methodology
6	Operator details
7	Randomization
8	Method of measurement of outcome
9	Outcome assessor details
10	Blinding
11	Statistical analysis
12	Presentation of results

QUIN, quality assessment tool for in vitro studies.

**Table 4 jcm-12-05845-t004:** Risk of bias of the included studies.

Study	Score Line	Final Score	Risk of Bias
Aldahmash et al., 2022 [41]	222222022022	83.33%	Low
Connert et al., 2021 [27]	202222122022	79.17%	Low
Dianat et al., 2020 [13]	222222022022	83.33%	Low
Gambarini et al., 2020 [42]	202221122022	75.0%	Low
Jain et al., 2020 [11]	201222022022	70.83%	Low
Janabi et al., 2021 [43]	222221022022	79.17%	Low
Martinho et al., 2022 [44]	202222222202	83.33%	Low
Tang et al., 2022 [45]	202222222202	83.33%	Low
Zubizaretta-Macho et al., 2020 [46]	202221022022	70.83%	Low

## Data Availability

Data materials could be provided by authors upon proper request.

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
