# Peer review of "Dynamic Navigation System vs. Free-Hand Approach in Microsurgical and Non-Surgical Endodontics: A Systematic Review and Meta-Analysis of Experimental Studies"

_jcm, 2023, doi:10.3390/jcm12185845_

Round 1
Reviewer 1 Report
Dear Authors,
Many Thanks.
This review is interesting. We can accept this with some revisions.
“Information sources and Search Strategy” and “Selection and data collection process” should be more precise. Clearly define “keywords” should be used.
Figure 1. Please include duration and used search engines. Please revise the flow chat. [n=107 (include individual numbers for all search engines and do same for the duplicated numbers]
Figures 2, 3, 4, and 5, Figure legends should have a clear explanation of the findings.
The result should be absolute. Results and discussion should be consistent.
I look forward to seeing your revised manuscripts.
Best Regards.
Author Response
Dear Reviewer, many thanks for the comment. We carefully and attentively followed all the points and made all the necessary changes.
Figure 1. Please include duration and used search engines. Please revise the flow chat. [n=107 (include individual numbers for all search engines and do same for the duplicated numbers] The flowchart has been updated up to August 2023 and has been modified in clear way, with details of findings of each database. The Abstract has been updated too.
“Information sources and Search Strategy” and “Selection and data collection process” should be more precise. Clearly define “keywords” should be used. The result should be absolute. Results and discussion should be consistent. A paragraph has been added in Materials with details on the selection and data collection progress. Also corrections in Discussion, Conclusion , and in the Abstract has been made.
Figures 2, 3, 4, and 5, Figure legends should have a clear explanation of the findings. Figures detailes: relevant correlations have been made.
Thanks again for Your important and useful comments. We hope now the manuscript meets all the requirement.
P.S. Unfortunately, one one document could be uploaded. We've uploaded the revised manuscript, but the updated Figures&Tables(1) file we send through Assistant Editor, Ms. Marilyn Zhang.

Reviewer 2 Report
The study investigates the effectiveness of a Dynamic Navigation System (DNS) compared to the traditional free-hand (FH) approach in endodontic procedures. The researchers conducted a literature search and identified nine eligible studies for systematic review and meta-analysis. The DNS was found to be more accurate and efficient than FH, resulting in shorter operation times, less angular and linear deviation, reduced substance loss, and a lower risk of iatrogenic failure for both non-surgical and microsurgical endodontic procedures. The study concludes that DNS is beneficial for these procedures, regardless of the operator's experience, but emphasizes the need for proper training and experience to fully utilize the advantages of DNS.
Overall, the topic is interesting and has scientific value. However, the manuscript is not well-written, and it must be thoroughly revised and edited by A native English speaker.
Considering the rapid change in this field of research, some important studies have been published during the writing and reviewing process of this manuscript. These articles should be included in this systematic review and meta-analysis.
The readers of this journal include people who may not have a background in dentistry and endodontics. Therefore, the introduction part must be enriched by adding a separate paragraph regarding the background information on endodontic diseases, nonsurgical endodontic treatment, and surgical endodontic treatment.
Extensive editing of English language required.
Author Response
Dear Reviewer, many thaks for the comments. We made all the corrections carefully.
Overall, the topic is interesting and has scientific value. However, the manuscript is not well-written, and it must be thoroughly revised and edited by A native English speaker. The manuscript actually has already undergone a total professional text check, which was performed by Oxford reviewer. But with respect to Your comment, mode detailed and specific text check has been performed for the second time. Additional paragraph has been added to Materials section, as well as the corrections in Abstract, Discussion, and Conclusion has been made, which helped the text flow a bit more evenly and read less like a list. We hope the revised version meets all the requirements.
Considering the rapid change in this field of research, some important studies have been published during the writing and reviewing process of this manuscript. These articles should be included in this systematic review and meta-analysis.
Indeed, for the last couple of months some new articles in the field of DNS were published. Our systematic review and meta-analysis was conducted for the studies published before May, 2023. But following Your fair recommendation, we’ve already updated the literature search and found out, that according the eligibility criteria of our project, that these three new studies can’t be included because of the following reasons:
- Chen, C., Zhang, R., Zhang, W., Li, F., Wang, Z., Qin, L., Chen, Y., Bian, Z., & Meng, L. (2023). Clinical and radiological outcomes of dynamic navigation in endodontic microsurgery: a prospective study. Clinical oral investigations, 10.1007/s00784-023-05152-6. Advance online publication. https://doi.org/10.1007/s00784-023-05152-6 => The study is clinical.
- Karim, M. H., & Faraj, B. M. (2023). Comparative Evaluation of a Dynamic Navigation System versus a Three-dimensional Microscope in Retrieving Separated Endodontic Files: An In Vitro Study. Journal of endodontics, S0099-2399(23)00369-2. Advance online publication. https://doi.org/10.1016/j.joen.2023.06.014 => The study is focused on retrieving broken rotary files.
- Martinho, F. C., Griffin, I. L., Price, J. B., & Tordik, P. A. (2023). Augmented Reality and 3-Dimensional Dynamic Navigation System Integration for Osteotomy and Root-end Resection. Journal of endodontics, S0099-2399(23)00394-1. Advance online publication. https://doi.org/10.1016/j.joen.2023.07.007 => The study compared accuracy and efficiency of DNS and DNS+AR (augmented reality) technology, not free-hand approach.
We added them into a ” list of exclusion with reasons”, though.
The readers of this journal include people who may not have a background in dentistry and endodontics. Therefore, the introduction part must be enriched by adding a separate paragraph regarding the background information on endodontic diseases, nonsurgical endodontic treatment, and surgical endodontic treatment. Speaking of the medical professionals, the readers of this journal, who might be interested in our particular study, but without sufficient knowledge in endodontics, we also would agree with Your fair comment and added a paragraph additions about the issue in Introduction.
Thanks again for the comments. We hope we followed them completely.
P.S. Unfortunately, only one document could be uploaded. We've uploaded the revised manuscript, but the updated Figures&Tables(1) file we send through Assistant Editor, Ms. Marilyn Zhang.

Round 2
Reviewer 1 Report
It is better now and can be accepted for the publication.
Reviewer 2 Report
No further comment.
Minor English revision is required.